# Short-Term Optimal Operation of Baluchaung II Hydropower Plant in Myanmar

**Jiqing Li [1], May Myat Moe Saw [1,2,*], Siyu Chen [1] and Hongjie Yu [3]**

[1]  School of Renewable Energy, North China Electric Power University, Beijing 102206, China;
    jqli6688@163.com (J.L.); c869783554@163.com (S.C.)
[2]  Baluchaung II Hydropower Station, Loikaw 09011, Myanmar
[3]  Zhejiang Design Institute of Water Conservancy and Hydroelectric Power, Hangzhou 310002, China;
    yhjyeah@163.com
[*]  Correspondence: ms.maymyatmoesaw@gmail.com or 1154300017@ncepu.edu.cn

**Abstract:** The short-term optimal operation model discussed in this paper uses the 2016 to 2018 daily and monthly data of Baluchaung II hydropower station to optimize power generation by minimizing water consumption effectively in order to get more revenue from optimal operation. In the first stage, run-off-river type Baluchaung II hydropower station data was applied in a mathematical model of equal micro-increment rate method for optimal hydropower generation flow distribution unit results. In the second stage, dynamic programming was used to get optimal hydropower generation unit distribution results. The resultant data indicated that optimized results can effectively guide the actual operation run of this power station. The purpose of the optimal load dispatching unit was to consider the optimal power of each unit for financial profit and numerical programming on the actual data of Baluchaung II hydropower plant to confirm that our methods are able to find good optimal solutions which satisfy the objective values of 17.75% in flow distribution units and 24.16% in load distribution units.

**Keywords:** short-term optimal operation; run-off-river; Baluchaung II hydropower station; equal micro-increment rate method; dynamic programming

## 1. Introduction

Water is used for many purposes such as drinking, cleaning, irrigation, power generation, recreation, and navigation. It is vital for all living things and also for infrastructure, industrialization, and urbanization of the country. In recent years, global warming has threatened the ecosystems of the earth that affects the daily obtainable water resources. If it is handled well with better technologies, flood and drought can be better controlled. Optimal water usage is becoming essential to protect water shortages in the dry season particularly in arid and semi-arid regions because obtainability of enough water resources and cost effectiveness of water usage impacts the management of hydropower generation. Hydropower generation is mostly affected by physical and economic factors. Physical factors depend on high flow rate from reservoirs, waterfalls and lakes, as well as dam reinforcement, flexible weather conditions, and silt-free water. Meanwhile, economic factors, power demand, capital investment, and lack of energy sources are the main factors for better power generation. Reckoning the costs of operating numbers of turbines, dams, intake, pipelines, fore bays, penstocks, powerhouses, protection, regulation, control, network connections, and transmission lines also supports the economic profit of hydropower plant [1].

The optimal operation of a hydropower station is to generate load with many constraints for many purposes [2]. There are many algorithms for mathematical models to optimize operation with

correctly conceptualized theories and formulas [3]. In designing a model, there are two methods used: optimization and simulation [4–6]. A simulation model is a descriptive model that tries to express the actual system of operation characteristics by using the complicated interrelations between components. Even though the model is easy and flexible to use, it needs much experimentation in hydropower-related applications. An optimization model is a normative model which attempt to get the utmost finest solution with the multiple suitable constraints. In fitting the structure and format of mathematical algorithm, these models have more limitations than simulation models.

There are three types of period to optimize the operation: short term which schedules for one to two weeks, medium term that programs for 3 to 18 months and long term for one to five years. The main goal of optimization of hydropower station is in order to meet the requirements of relevant departments of national economy and society according to above-mentioned principles of operation with certain optimization theories and methods. This means to optimize the power generation in order to get stable and economical in load balancing with the utmost total direct and indirect benefits. Direct benefit is concerned with the primary operation benefit got from the power generation operation dispatching for the national load demand. Indirect benefits relates to the social and ecological impacts got from controlling the power generation, navigation, irrigation, flood control, and urban water supply.

Therefore, the reason for conducting short-term optimal operation of Baluchaung II hydropower station in Myanmar is to optimize the power generation by minimizing water consumption and evaluate the optimal flow and load distribution units by equal micro-increment rate method and dynamic programming. The calculation data are from the Baluchaung II hydropower station in Myanmar with the installed capacity of 168 MW (28 MW × 6). It is a run-off-river type hydropower station that supplies the national electricity demand as a base load power station in Myanmar. Unfortunately, although there are plenty of papers about hydropower status reports, there is a few technical papers about optimal operation relating to reservoirs and flood control in Myanmar. Therefore, this paper can be helpful to manage the hydropower plant for optimal operation. Many scholars have researched how to use water efficiently in optimal design, planning, and operation in economics and engineering fields.

The application of the researched methods depends on the characteristics of the different types of hydropower stations. Methods are classified into two main groups: heuristic and mathematical programming. Heuristic methods include improving particle swarm optimization [7–9], progressive optimal algorithms [10,11], improved progressive optimal algorithm [12], chaos cultural sine cosine algorithm [13], and recursive optimization [14]. Mathematical programming includes linear programming [15], mixed integer linear programming [16], stochastic programming, and dynamic programming. Hybrid algorithms give better results and are more accurate than single algorithms and the key limitation of them is that they cannot improve the results very much more than actual operation.

From the above literature, it can be noted that the purpose of short-term optimal operation is to regulate the water flow hour-by-hour in order to stabilize the scheduling periods for the peak hours of the day and forecast the daily run-off between the conditions of multi-constraints in a complex system. Due to complex multi-constraints between water level and output in modelling of hydro power stations, feasibility of scheduling methods is still needed to improve on practical operations, and more new hybrid algorithms need to be considered for future optimization issues. Also, in actual power generation, the load demand changes need to be taken into account because they are a main factor of the instability of load distribution units. Some research papers give general conclusions and do not clearly point out the advantages and the differences between the results. In short-term load dispatching problems, dynamic programming is a better method because of steady computational processes in finding optimal solutions. Therefore, dynamic programming is used in this paper in order to get better optimal results in the case of load swinging.

## 2. Case Study

### 2.1. Baluchaung II Hydropower Plant

Myanmar is a Southeast Asian country which depends mainly on its water resources for electricity supply. Although it is implementing many hydropower stations for the country's needs, problems of electricity shortage are still encountered during the summer season. Therefore, maximum load with minimum water usage is the aim for hydropower plants in Myanmar in order to optimize the load distribution of the generator units. Most of the hydropower plants in Myanmar are the reservoir type and some are run-off river type. They are essential to produce electricity which requires no fuel and are much simpler to operate and maintain than other types of power station due to lower operating costs. The total cost of run-off river power plants mainly rely on the number of turbines [17]. The power capacity of a hydropower plant is primarily the function of two variables: flow rate expressed in cubic meters per second and the hydraulic head, which is the elevation distance the water falls in passing through the power plant. A typical hydropower plant can be classified according to rated power capacity, type of turbine, water head, and location and type of dam as well as sizes which includes large, small, micro, and pico, depending on generated MW. The basic components of hydropower plants are intake, dam, headrace, fore bay, penstock, power house, turbine, generator, and tailrace. Baluchaung II hydroelectric power station is located in Loikaw which is the Southern Shan state and it generates power from the Baluchaung River, a tributary of the Thanlwin River, which is located in the Middle East region of Myanmar. The water enters to the Baluchaung River from the Mobye Reservoir which collect water from the Inle Lake. The construction of the Baluchaung II power station was divided into two phases. Investigations were started in 1954 and construction was started in 1960 for the first three generator (28 MW × 3) as a Japanese post-war reparation project. The second-phase (28 MW × 3) followed in the period between 1970 and 1974 and was self-funded. The power house has six units of turbine generators. Units 1 to 3 were constructed during the first stage construction and units 4 to 6 were constructed as the second stage. All turbines are horizontal Pelton type turbines that have two runners with twenty buckets on the both side of the generator combined by the main shaft and the each runner receives a water jet from two nozzles and has two horizontal shaft rotary inlet valves. The maximum capacity is 168 MW, maximum discharge is 47.58 m$^3$/s, total head is 436.6 m and effective head is 416.5 m. The generated power is transported to the Yangon and Mandalay regions via 230 kV and 132 kV transmission lines. The power station has been supplying more than 50% of the total electric power of Myanmar for the past 40 years. The facility capacity and the output of electric power have been 20% and 40%, respectively, of the total in Myanmar in last two decades and therefore, it was very important for country's electricity needs because it was the first large-scale hydropower plant established and it is always considered as the base load power plant especially in dry season. The Ministry of Electricity and Energy aims to implement a Myanmar electricity master plan to generate 100 percent electricity with the energy generation mix of 5% solar and wind, 8% natural gas, 30% coal and hydropower more than half of electricity in 2030. The national economy depends on electricity, peace, and human resources. Therefore, trying to get access electricity for the 22 million population is the country's goal as a developing country. The optimal operation of hydropower plant and waterway system of Baluchaung II hydropower station is described in the Figure 1.

### 2.2. Methodology

The methodology includes three parts. The first part introduces the optimal operation of hydropower plant. The second part is about the equal micro-increment rate method and the third part explains the dynamic programming method. The equal micro-increment rate method was used for programming the results of optimal hydropower generation flow and the dynamic programming method was used for hydropower generation. The objective of this paper is to optimize the hydropower station in the unit and hydropower generation level and to upgrade the efficiency in the higher stage.

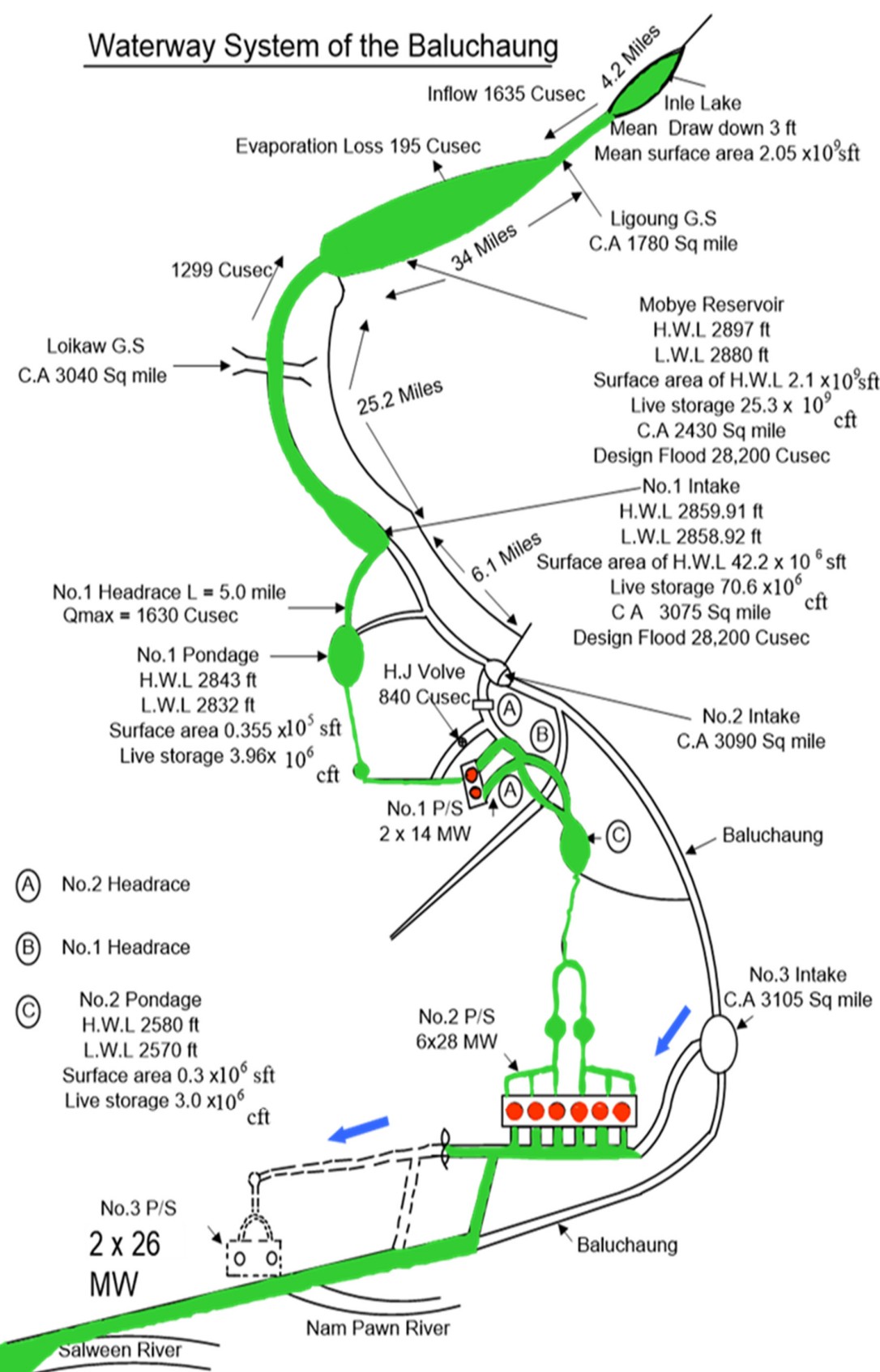

**Figure 1.** Waterway System of Baluchaung II hydropower station.

Establishment of the Optimal Operation Model in the Plant

The basic task of the optimal operation of the hydropower station is to rationally arrange the load distribution of the units [18,19]. These units are put into operation on the basis of the known power generation flow or output of the hydropower station to achieve the highest power generation efficiency of the whole plant and increase the economic benefits of the hydropower station. The following principles should be followed when conducting research on the optimal operation of the plant.

(1) Optimization principle.

The economic operation of the plant can adopt different economic principles according to different situations. When the power generation flow of the power station is given in the "water to electricity" mode, the power station maximizes its output by optimizing the unit commitment and water distribution. When the load of the power station is given in the "electricity to water" mode, the power station minimizes its power generation water consumption by optimizing the unit commitment and load distribution.

(2) Principles for safe operation of power stations and units.

When the unit is running in certain output and head areas, severe vibrations will occur. This is called the vibration zone. Vibration is a resonance phenomenon caused by mechanical, hydraulic, and electromagnetic vibrations, which seriously threaten the safe production of the power station and the service life of the unit. Therefore, the optimal operation of the plant should avoid the vibration zone operation.

According to the actual production of "electricity to water" mode by Baluchaung II hydropower station, this paper uses the minimum water consumption as the optimization criterion to establish the optimal operation model of the plant. The objective function is as follows:

$$Q = \min \sum_{i=1}^{n} Q_i(N_i, H) \tag{1}$$

where $Q$ is the total power generation flow of the hydropower station, $i$ is the unit number, $N_i$ is the output of the unit $I$, $H$ is the water head, $Q_i(N_i, H)$ is the power generation flow when the unit $N_i$ is output, and the head is $H$.

The constraint condition is the balance of output force. Equation (2) explains the constraints of power output. It should be noted that after many years of operation, the Baluchaung II hydropower station is in good running condition and there is no serious vibration of the unit. Therefore, the model does not consider the vibration zone constraint.

$$\begin{cases} \sum\limits_{i=1}^{n} N_i = N \\ N_{\min}(H) \leq N_i \leq N_{\max}(H) \end{cases} \tag{2}$$

In the formula, $N_{\min}(H)$ and $N_{\max}(H)$ are the lower limit and upper limit of the output of unit $i$ when the head is $H$.

*2.3. Equal Micro-Increment Rate Method for Model Solving*

Hydropower station economic operation model-solving methods are generally divided into two categories. One is the traditional method, which mainly refers to the graphical method using the flow increase rate characteristic curve and the flow characteristic curve, the most representative method of equal micro-increment rate. The other is mathematical optimization methods, mainly other modern mathematical methods such as dynamic programming. The principle of the micro-increment rate method is easy and the calculation is simple. The flow characteristics of the unit of Baluchaung II

hydropower station that meet the requirements of the micro-increment rate method must be in the condition of a smooth and convex curve. Therefore, the method of solving for the Baluchaung II hydropower station is the equal-increment rate method economic operation model and dynamic programming. When solving the objective function corresponding to Equation (3), the Lagrangian function $F$ can be constructed:

$$F = \sum_{i=1}^{n} Q_i + \lambda(N - \sum_{i=1}^{n} N_i) \tag{3}$$

The necessary conditions for $F$ to reach the extreme point are:

$$\begin{cases} \frac{\partial F}{\partial N_1} = \frac{\partial Q_1}{\partial N_1} - \lambda = 0 \\ \frac{\partial F}{\partial N_2} = \frac{\partial Q_2}{\partial N_2} - \lambda = 0 \\ \qquad \vdots \\ \frac{\partial F}{\partial N_n} = \frac{\partial Q_n}{\partial N_n} - \lambda = 0 \end{cases} \tag{4}$$

To express the flow rate of micro-increment of the $i$-th unit when the unit is fixed, Equation (5) can be expressed as:

$$q_1 = q_2 = q_3 = q_n = \frac{1}{\lambda} = constant \tag{5}$$

Equation (5) is the principle of the micro-increment rate of the optimal flow distribution between the operating units. When the unit models in the hydropower station are the same, that is, the micro-increment rate curves of the units are the same, the calculation result of the micro-increment rate method is the average load of the power station which is evenly distributed to each unit that is put into operation. At this time, the key to the optimal operation of the plant is to determine the number of units that are put into operation. Therefore, when solving the economic operation model of the Baluchaung II hydropower station when the total output of the power station is certain, one to six units of known data are required to calculate the situation and select the scheme with the smallest total power generation flow.

### 2.4. Dynamic Programming

Dynamic Programming (DP) is a traditional method like linear and non-linear programming introduced by Richard Bellman in 1950s for multiple purposes like applications for economics, engineering, and also in the army [20]. It is used not only in mathematics but also in computer programming. To calculate successfully, the order of the sequence is from the beginning state to final state. The optimization principle can be expressed as the optimal decision sequence from the initial state and initial decision of the process to the next decision state. Well known DP algorithms are Unix diff for comparing two files, Bellman-Ford for shortest path routing in networks, and TeX, the ancestor of LaTeX for score predictor. The advantage of DP is getting final results by memorization to avoid repetitive work. Two types of DP problems are optimization and combinatorial. These can used in two approaches: top-down and bottom-up approaches. The calculation steps of DP are establishment of dynamic programming recursion equation, recursive calculation, and real-time economic operation plan formulation. These are shown below.

#### 2.4.1. Stage and Stage Variables

For real-time economic operation problems, each unit can be staged and put into the unit. The number $i$ represents the phase variable ($i$ = 1, 2, ... , $n$). $i$ is the unit stage, and 1~($i - 1$) is the remaining period.

### 2.4.2. State Variables

State variables are the total output of 1~*i* units $P_i \sum(t)$ $(i = 0, 1, \ldots, n)$ where $P_o \sum(t) = 0$ as the *i*-th order selecting and it needs to describe the evolution of the process with no after-effects.

### 2.4.3. Decision Variables

Decision variables are the output $P_i(t)$ of each unit, and the t-time (segment) of the *i*-th unit. If the force range composition allows the decision set $D_i(t)$, then $P_i(t) \in D_i(t)$. The sub-strategy is recorded as: $U_i = \{P_1(t), P_2(t) \ldots, P_i(t)\}$.

### 2.4.4. State Transition Equation

In the state transition equation the any stage (unit) *i*, describes the end state $P_i \sum(t)$ with stage initial state $P(i-1) \sum(t)$ for any stage unit *i*. The mathematical relationship between the decision and the decision $P_i(t)$ is called the state transition equation, and the load balance equation of the model is the state transition

$$P_i \sum(t) = P_{i-1} \sum(t) + P_i(t) \tag{6}$$

For this deterministic decision process, the state of the next phase is completely determined by the state and decision of the time period.

### 2.4.5. Index Function and Optimal Value Function

In its function, the power flow $Q_i(P_i(t))$ of the *i*-th stage represents the index function, 1~*i*. The optimal value of the total power generation flow in the stage $Q_i^*((P_i \sum(t))$ represents the optimal value function.

$$Q_i(P_i(t)) = Q_i(P_i(t), H(t)) \tag{7}$$

$$Q_i^* (P_i \sum(t)) = min_{ui}\left\{\sum_{j=1}^{i} Qi(Pj(t))\right\} \tag{8}$$

### 2.4.6. Recursive Equation

According to the multi-stage decision principle and Equations (7) and (8), the following sequence recursion can be listed

$$\begin{cases} Q_i^*(P_i \sum(t) \\ Q_o^*(0) = 0 \end{cases} = \begin{matrix} min \\ P_i(t) \in D_i(t) \end{matrix} \{Q_i(P_i(t), H(t)) + Q_{i-1}^* (P_i \sum(t) - P_i(t))\} \begin{matrix} (i = 1 \sim n) \\ (i = 0) \end{matrix} \tag{9}$$

### 2.4.7. Constraints

The constraint in the recursive calculation is mainly the unit output limitation of Equations (10) and (11) which are used in the back-calculation to determine the optimal decision.

$$(P_i{}^{min}(t) \leq (P_i(t)) \leq P_i{}^{max}(t)) \tag{10}$$

$$\sum_{i=1}^{n} P_i(t) = P_s(t) \tag{11}$$

## 3. Calculation and Discussion

In the calculation steps, Microsoft excel was used to sort out the average power generation flow rate and generation output for each month from the 2016 to 2018 data. Then, the unit characteristics curve was fixed by exponential curves and the unit dynamic characteristic curve compared with other power plant performance curves such as the upstream water level capacity curve, and downstream

water level flow curve [21]. C# software was used to write the program for optimal generation flow and actual generation flow from the known data of six units from Baluchaung II hydropower station. The equal micro-increment rate method was used for calculation. The dynamic programming method was applied for optimal and actual power generation and the results compared to get the better optimal solution.

### 3.1. Unit Characteristic Curve Fitting

The Baluchaung II hydropower station is a run-off-river power station. This is why the upstream water level and the downstream water level of the power station remain unchanged and the power head is constant at 1388 feet. Therefore, the upstream water level storage capacity curve and the downstream water level flow curve are not required for the model solution. The unit dynamic characteristic curve which is the NQ curve (which should have been the NQH curve, but reduced to the NQ curve since H has been determined to constant head) is required to find out the optimal operation [22]. The Baluchaung II hydropower station has six identical units, and its unit characteristic curve is basically the same, but due to the reasons that may not be provided by the manufacturer, or the fact that the theoretical curve may not match the actual operation, there is no suitable model for solving the problem of unit dynamic characteristic curve. Therefore, this paper uses the measured data in the historical running process to perform curve fitting. Table 1 shows the measured data of Baluchaung II hydropower station.

**Table 1.** Measured data of Baluchaung II hydropower station.

| Date | Q | Unit | | | | | | E | Average E | Total | Net Total Transmission |
|------|---|------|---|---|---|---|---|---|-----------|-------|------------------------|
| | | 1 | 2 | 3 | 4 | 5 | 6 | | | | |
| 1.Mar.2016 | 1202 | 599 | 584 | 591 | 596 | 607 | 602 | 3579 | 149 | 3555 | 3550 |
| 2.Mar.2016 | 1201 | 542 | 567 | 567 | 579 | 569 | 536 | 3361 | 140 | 3335 | 3331 |
| 3.Mar.2016 | 1102 | 504 | 529 | 537 | 541 | 550 | 540 | 3201 | 133 | 3177 | 3173 |
| 4.Mar.2016 | 1202 | 488 | 485 | 494 | 541 | 508 | 490 | 3005 | 125 | 2982 | 2978 |
| 5.Mar.2016 | 1202 | 503 | 480 | 494 | 498 | 510 | 464 | 2948 | 123 | 2926 | 2921 |
| 6.Mar.2016 | 1202 | 431 | 402 | 456 | 421 | 468 | 481 | 2859 | 111 | 2636 | 2634 |
| 7.Mar.2016 | 1203 | 513 | 438 | 508 | 477 | 512 | 548 | 3004 | 125 | 2981 | 2976 |
| 8.Mar.2016 | 1104 | 473 | 476 | 471 | 485 | 467 | 475 | 2847 | 119 | 2825 | 2820 |
| 9.Mar.2016 | 1105 | 485 | 450 | 497 | 497 | 507 | 479 | 2916 | 121 | 2889 | 2885 |
| 10.Mar.2016 | 1102 | 461 | 4545 | 435 | 4763 | 448 | 451 | 2726 | 114 | 2705 | 2701 |

Since the hydropower station runs more than one unit, when the output of each unit is different, it is impossible to determine the corresponding flow rate of each unit under its output. Therefore, in the curve fitting, it is necessary to select a period in which the unit output is relatively close and one day is the time period in this article, such as 27 September 2016. For the selected day, the total output of the hydropower station and the total power generation flow are evenly distributed to each unit as the basic data of curve fitting. From 2016 to 2018, the time period selected meets the requirements and form the basic data set. The basic data set was fitted in logarithmic form, and the curve obtained is shown in Figure 2. The NQ curve relationship was obtained as follows:

$$N = 9425.5 \times \exp(0.1314Q) \tag{12}$$

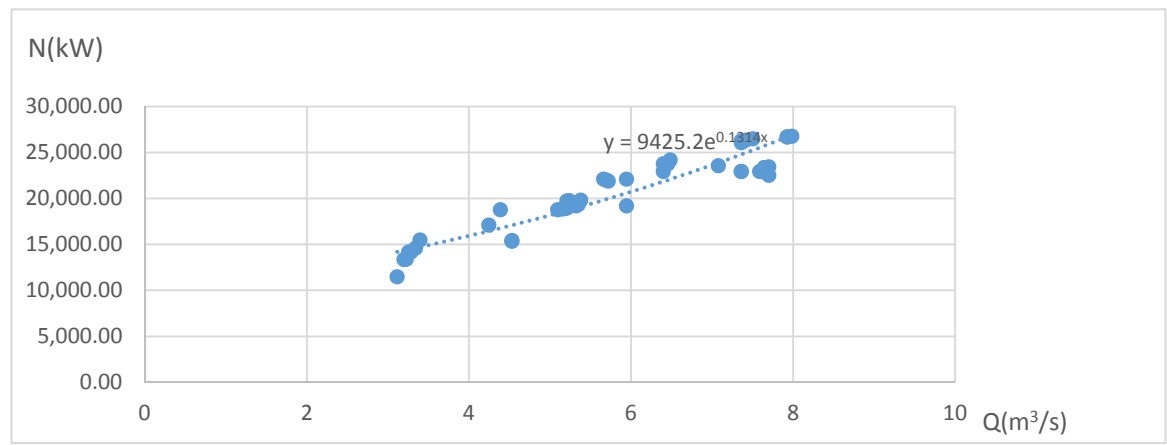

**Figure 2.** Unit dynamic characteristic NQ curve.

After fitting the curve, the equal micro-increment method was used to calculate the results of optimal power generation flow and dynamic programming was used to find the optimal power generation. Using the data from 2016 to 2018, the fitting accuracy test was carried out, that is, the power generation flow of each unit was obtained by inversely calculating the power generation flow according to the measured output of each unit, and then the total power generation flow of the hydropower station was calculated, shown in Table 2, and the measured power generation flow rate of the hydropower station was performed. The results are shown in following figures and tables.

**Table 2.** Power generation flow results of Baluchaung II hydropower station.

| Months | 2016 | | | 2017 | | | 2018 | | |
|---|---|---|---|---|---|---|---|---|---|
| | $Q_a$ (m³/s) | $Q_o$ (m³/s) | Efficiency (%) | $Q_a$ (m³/s) | $Q_o$ (m³/s) | Efficiency (%) | $Q_a$ (m³/s) | $Q_o$ (m³/s) | Efficiency (%) |
| Jan | 944 | 845 | 11% | 1935 | 1774 | 8% | 2011 | 1910 | 5% |
| Feb | 1085 | 974 | 11% | 1973 | 1835 | 7% | 2079 | 2018 | 3% |
| Mar | 1834 | 1586 | 13% | 1985 | 1854 | 7% | 2070 | 2005 | 3% |
| Apr | 1584 | 1318 | 17% | 1973 | 1850 | 6% | 2051 | 2023 | 2% |
| May | 1876 | 1641 | 12% | 2088 | 2031 | 3% | 2157 | 2144 | 1% |
| June | 1757 | 1522 | 14% | 1718 | 1604 | 7% | 2017 | 1913 | 5% |
| Jul | 1466 | 1245 | 15% | 1446 | 1294 | 11% | 2122 | 2088 | 2% |
| Aug | 1098 | 959 | 12% | 1527 | 1339 | 13% | 2047 | 1955 | 5% |
| Sep | 468 | 398 | 14% | 1001 | 857 | 15% | 1832 | 1624 | 11% |
| Oct | 1115 | 963 | 15% | 1522 | 1372 | 12% | 1787 | 1598 | 11% |
| Nov | 1058 | 870 | 18% | 2053 | 1979 | 4% | 2121 | 2085 | 2% |
| Dec | 1498 | 1274 | 16% | 2105 | 2064 | 2% | 2103 | 2082 | 1% |

*3.2. Water Flow Optimization*

In the Table 2, $Q_a$ means the actual power generation flow and $Q_o$ means the optimal power generation flow. In contrast, the maximum efficiency of power generation, in November 2016, was 18 percent whereas the lowest, in May 2018, was 1 percent, when the optimal power generation flow was highest at 2144 m³/s. It can be noted that efficiency is higher when the power generation flow is lower. The optimal solution can save over 200 m³/s water in some months whereas in some other literature, the optimized solution could save, at the most, 10 m³/s [6].

In Table 3, the actual solution consumes water discharge of 1662 m³/s with six distribution units whereas the optimal solution used water release of 1396 m³/s with four distribution units on 12 March 2016, as highlighted in red. The optimal solution, saved two turbines of water consumption and released a minimum discharge of water. From the calculated results, it can be seen that the lower the water consumption, the more cost-effective the power generation. Therefore, the equal micro-increment

rate method gave the best solution for optimal power generation flow, saved at least one turbine of consumption during most days, and consumed less water than the actual operation every day during the three years.

**Table 3.** The power generation flow distribution of units in actual and optimal operations.

| Date | Actual Power Generation Flow (m³/s) | | | | | | Optimal Power Generation Flow (m³/s) | | | | | |
|---|---|---|---|---|---|---|---|---|---|---|---|---|
| | Unit 1 | Unit 2 | Unit 3 | Unit 4 | Unit 5 | Unit 6 | Unit 1 | Unit 2 | Unit 3 | Unit 4 | Unit 5 | Unit 6 |
| 1.Mar. 2016 | 348 | 343 | 346 | 348 | 351 | 349 | 371 | 371 | 371 | 371 | 371 | 183 |
| 2.Mar. 2016 | 329 | 338 | 338 | 342 | 339 | 327 | 371 | 371 | 371 | 371 | 371 | 3 |
| 3.Mar. 2016 | 316 | 325 | 327 | 329 | 332 | 329 | 371 | 371 | 371 | 371 | 319 | - |
| 4.Mar. 2016 | 310 | 308 | 312 | 329 | 317 | 310 | 371 | 371 | 371 | 371 | 236 | - |
| 5.Mar. 2016 | 315 | 307 | 312 | 313 | 318 | 300 | 371 | 371 | 371 | 371 | 207 | - |
| 6.Mar. 2016 | 287 | 275 | 297 | 283 | 302 | 307 | 371 | 371 | 371 | 371 | 152 | - |
| 7.Mar. 2016 | 319 | 290 | 317 | 305 | 319 | 331 | 371 | 371 | 371 | 371 | 236 | - |
| 8.Mar. 2016 | 304 | 305 | 303 | 309 | 302 | 305 | 371 | 371 | 371 | 371 | 144 | - |
| 9.Mar. 2016 | 309 | 295 | 313 | 313 | 317 | 306 | 371 | 371 | 371 | 371 | 188 | - |
| 10.Mar. 2016 | 299 | 820 | 289 | 832 | 294 | 295 | 371 | 371 | 371 | 371 | 43 | - |
| 11.Mar. 2016 | 302 | 297 | 300 | 304 | 308 | 301 | 371 | 371 | 371 | 371 | 115 | - |
| 12.Mar. 2016 | 276 | 276 | 271 | 280 | 275 | 284 | 371 | 371 | 371 | 283 | / | - |
| 13.Mar.2016 | 299 | 289 | 293 | 301 | 298 | 291 | 371 | 371 | 371 | 371 | 17 | - |
| 14.Mar.2016 | 303 | 294 | 294 | 303 | 271 | 300 | 371 | 371 | 371 | 371 | 6 | - |
| 15.Mar.2016 | 293 | 287 | 293 | 289 | 313 | 297 | 371 | 371 | 371 | 371 | 27 | - |

The generation flow curves for 2016 to 2018 are shown in Figure 3. From the calculated results, the solid line curve corresponds to the actual hydropower generation flow, whereas the dotted line curve corresponds to the optimal hydropower generation flow. In November 2016, as shown in Figure 3a, the two curves fluctuated together with a small gap of between 800 m³/s and 1100 m³/s. This means that the two curves were synchronized until the end of the month with the actual power generation curve was higher than the optimal power generation flow showing the use of less water in optimal rather than actual operation and denoting that it was optimal operation. The red dotted rectangular shape shows that water consumption reach its lowest point in November 19 which is 718 m³/s in optimal operation and 949 m³/s in actual operation with the efficiency of 24.3 percent. In Figure 3b, the actual operation curve is the same with optimal operation because both unit generations were similar to each other in May 2018. However, on only one day in May 2018 it dropped to minimum value, where the blue dotted circle which reaches 1791 m³/s in optimal and 1945 m³/s in actual solution and efficiency to 7.9 percent. By comparing two months in Figure 4, it can be seen that the smaller the water flow gets the higher the efficiency in order to get more benefit in power production. From the above calculated curves, it can be noted that the volumetric flow rate cannot be constant as it is changing in each period of time and the results show the optimal solution curves are always lower than actual solution curves because of the optimal resulted values are lower than actual solution to minimize water consumption in order to get more revenue from power generation.

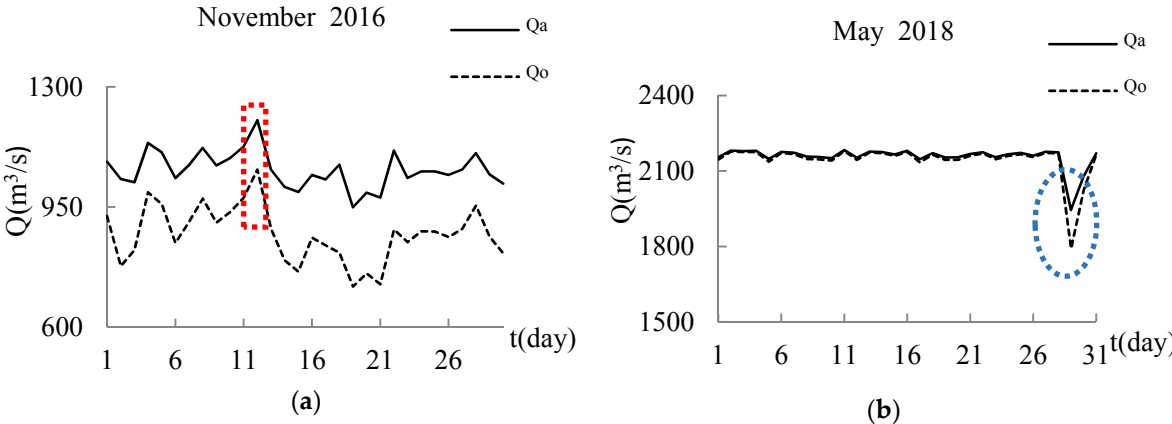

**Figure 3.** Actual and optimal power generation flow chart for November and January 2016.

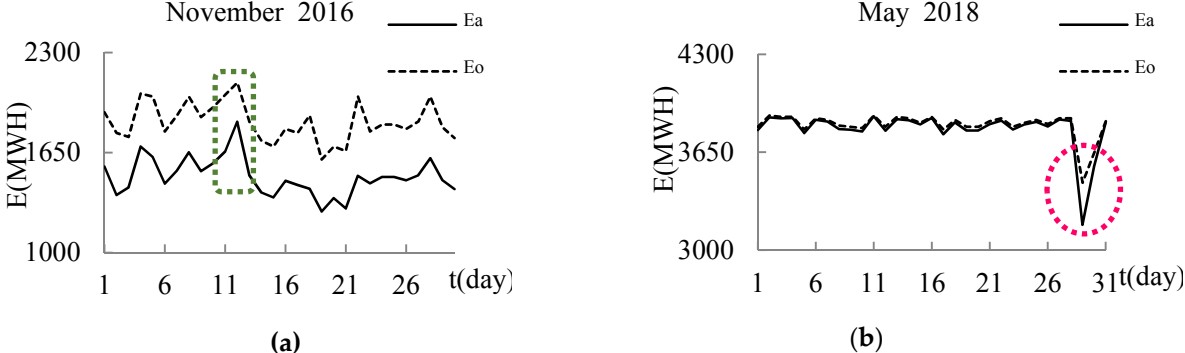

**Figure 4.** Optimal and actual power generation chart of January 2016 and February 2017.3.3. Power Generation Optimization.

In the Table 4, $E_a$ means the actual power generation and $E_o$ means the optimal power generation. From the calculated results of three years, the maximum efficiency of power generation was 24.16 percent and minimum efficiency was 0.72 percent. The lowest optimal power generation was 1851.7 MWH and the highest optimal power generation was 3831.9 MWH which is the full load of five units.

**Table 4.** Power generation results of Baluchaung II hydropower station.

| Month | 2016 | | | 2017 | | | 2018 | | |
|---|---|---|---|---|---|---|---|---|---|
| | $E_o$ | $E_a$ | Efficiency | $E_o$ | $E_a$ | Efficiency | $E_o$ | $E_a$ | Efficiency |
| Jan | 1602 | 1455 | 10% | 3426 | 3145 | 9% | 3547 | 3384 | 5% |
| Feb | 1873 | 1678 | 15% | 3493 | 3251 | 8% | 3669 | 3566 | 3% |
| Mar | 3083 | 2793 | 11% | 3511 | 3289 | 7% | 3636 | 3532 | 3% |
| Apr | 2783 | 2310 | 20% | 3488 | 3271 | 8% | 3625 | 3573 | 2% |
| May | 3295 | 2972 | 11% | 3689 | 3600 | 3% | 3832 | 3805 | 1% |
| June | 3099 | 2677 | 19% | 3029 | 2816 | 9% | 3564 | 3383 | 6% |
| Jul | 2574 | 2181 | 19% | 2533 | 2275 | 13% | 3764 | 3699 | 2% |
| Aug | 1898 | 1650 | 15% | 2683 | 2356 | 15% | 3598 | 3466 | 4% |
| Sep | 801 | 684 | 10% | 1734 | 1494 | 16% | 3240 | 2872 | 13% |
| Oct | 1934 | 1679 | 17% | 2674 | 2405 | 15% | 3162 | 2814 | 14% |
| Nov | 1852 | 1493 | 24% | 3518 | 3512 | 1% | 3780 | 3715 | 2% |
| Dec | 2629 | 2233 | 20% | 3718 | 3644 | 2% | 3728 | 3686 | 1% |

The dynamic programming algorithm bottom-up approach method was used in order to produce the optimal distribution of units which are shown in Table 5. Based on actual load distribution of units,

the optimal outcomes obtained with 0.1 MWH discrete interval and vibration area does not take into consideration for the adjustment of unit distribution. It can be seen that the actual load distributes six units with 3276.3 MWH and the optimal solution get 3351 MWH which distributes five units and four units are full load which are highlighted in red color.

**Table 5.** The load distribution of units in actual and optimal operations.

| Date | Actual Power Generation (MWH) | | | | | | Optimal Power Generation (MWH) | | | | | |
|---|---|---|---|---|---|---|---|---|---|---|---|---|
| | Unit 1 | Unit 2 | Unit 3 | Unit 4 | Unit 5 | Unit 6 | Unit 1 | Unit 2 | Unit 3 | Unit 4 | Unit 5 | Unit 6 |
| 1.Dec.2018 | 639 | 638 | 641 | 658 | 660 | 657 | 672 | 672 | 672 | 672 | 672 | 545 |
| 2.Dec.2018 | 612 | 604 | 558 | 631 | 630 | 622 | 672 | 672 | 672 | 672 | 672 | 359 |
| 3.Dec.2018 | 652 | 643 | 645 | 664 | 667 | 664 | 672 | 672 | 672 | 672 | 672 | 576 |
| 4.Dec.2018 | 649 | 645 | 646 | 666 | 665 | 663 | 672 | 672 | 672 | 672 | 672 | 576 |
| 5.Dec.2018 | 640 | 637 | 640 | 655 | 654 | 647 | 672 | 672 | 672 | 672 | 672 | 525 |
| 6.Dec.2018 | 631 | 620 | 632 | 649 | 652 | 645 | 672 | 672 | 672 | 672 | 672 | 486 |
| 7.Dec.2018 | 626 | 615 | 636 | 650 | 652 | 646 | 672 | 672 | 672 | 672 | 672 | 486 |
| 8.Dec.2018 | 645 | 62 | 645 | 666 | 663 | 662 | 672 | 672 | 672 | 672 | 672 | 35 |
| 9.Dec.2018 | 642 | 19 | 632 | 665 | 661 | 658 | 672 | 672 | 672 | 672 | 663 | - |
| 10.Dec.2018 | 622 | 0 | 626 | 648 | 659 | 626 | 672 | 672 | 672 | 672 | 516 | - |
| 11.Dec.2018 | 655 | 110 | 655 | 666 | 669 | 661 | 672 | 672 | 672 | 672 | 672 | 88 |
| 12.Dec.2018 | 644 | 473 | 643 | 666 | 667 | 663 | 672 | 672 | 672 | 672 | 672 | 415 |
| 13.Dec.2018 | 562 | 579 | 638 | 670 | 666 | 666 | 672 | 672 | 672 | 672 | 672 | 441 |
| 14.Dec.2018 | 610 | 603 | 530 | 617 | 618 | 612 | 672 | 672 | 672 | 672 | 672 | 307 |
| 15.Dec.2018 | 639 | 637 | 646 | 555 | 660 | 656 | 672 | 672 | 672 | 672 | 672 | 450 |

As shown in Figure 4, the optimal power generation curve was higher than actual power generation curve in November, 2016. This means that the power generation was optimal and the cost was economical as it could be run effectively with less water consumption. The highest power generation was reached on 12 November, as shown in green dotted rectangular in Figure 4a, when optimal power generation was 2105 MWH whereas the actual power generation was 1850.4 MWH and the efficiency was 13.7 percent, the lowest among in this month. This means the power generation is optimal when the optimal value is larger than actual value whereas the efficiency is minimum. In May 2018, as shown in Figure 4b, the optimal and actual power generation curves overlap each other between 3800 MWH and 3900 MWH and drop sharply to 3488.3 MWH in 29 May, as shown in the pink dotted circle, therefore, when the optimized power generation was larger than the actual operating power generation result, and the optimization efficiency was low. In contrast, the actual and optimal power generation was the same and there was almost no gap within the two curves in most days.

From the data shown in Figure 5, the generation curves oscillate inconstantly due to the load demand changes that caused the load distribution units instability. In actual power generation, the emergency start up and shut down raises the water consumption level. In water consumption, the optimal solution saves 1.36% more than actual operation [23] whereas it saves up to 2% for the same turbine types [14]. In this paper, the optimal water consumption saved up to 10% more than actual operation and the load generation was higher than actual operation in optimal solution in every time period from 2016 to 2018. In contract, the results give the advantageous optimization operation which satisfies the cost benefit effects.

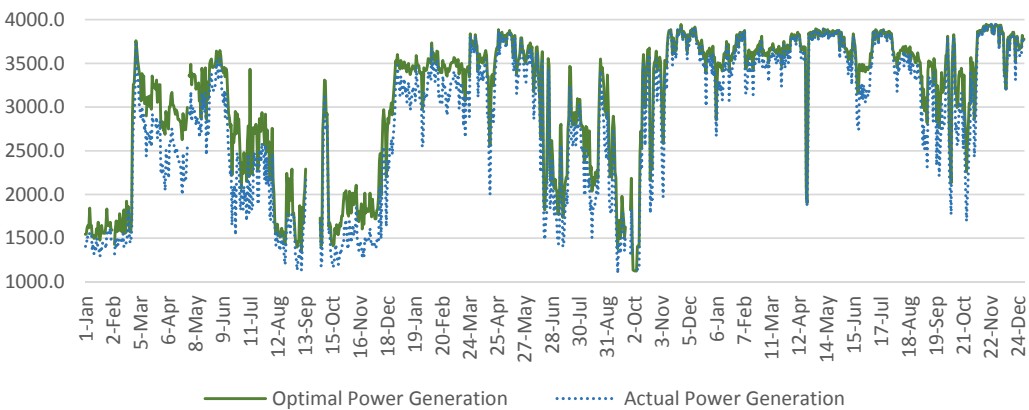

**Figure 5.** Optimal and power generation chart for 2016–2018.

## 4. Conclusions

By optimizing the optimal operation of hydropower station from the 2016 to 2018 daily and monthly data of Baluchaung II hydropower station by programming, the optimized power generation flow of the hydropower station could be obtained. Dynamic programming was used for optimal generation with minimum generation flow for saving water especially in the drought season. In power generation flow, the best optimization result was obtained when the optimization efficiency reached 17.75% when water quantity was small whereas in power generation, the optimization efficiency is 24.16% for better optimal operation. In the unit allocation, based on the characteristics of the fitted NQ curve, we found that maximizing the output of the unit could optimize the efficiency of the power generation flow and also noticed that the optimization effect of the power generation flow was poor due to the incoming water volume being large. However, in this simulation, the results showed that total water consumption of optimal solution could save a larger amount of water consumption than actual operation and monthly scale generation flow results for optimal solution were lower than the actual operation data in every month during three years. Whereas in power generation, the optimization result was higher than the actual result in each month within the three years. Due to the point in contrast, the operation was optimal within three years. The key limitation of this paper, is that it does not consider some other constraints in the reckoning process and the future scholar could take into account start up/shut down and power plant equipment constraints in the case of optimal operation.

**Author Contributions:** J.L. contributed conceptualization, methodologies, corrections to the paper and gave useful suggestions. M.M.M.S. and S.C., contributed to the data collection, calculation, made figures and tables and wrote the paper H.Y. programmed the C# software. All authors have read and agreed to the published version of the manuscript.

**Funding:** This study was financially supported by the National Key Projects of China "Water resources efficient development and utilization" (2016YFC0402208, 2017YFC0405900) and National Natural Science Foundation of China (No.51641901).

**Acknowledgments:** The authors would like to thank to the Baluchaung II hydropower station for giving data for the above projects and for financial support.

**Conflicts of Interest:** The authors declare no conflict of interest.

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
