# Peer review of "Short-Term Optimal Operation of Baluchaung II Hydropower Plant in Myanmar"

_water, doi:10.3390/w12020504_

Round 1

Reviewer 1 Report

This paper presents the short-term optimal operation of a  hydropower plant. The topic of the paper may be interesting, however, there are some points that must be cover as follows:

The Abstract in its current from is an alternative Introduction, it should clearly describe the scope with more focusing on the proposed approach and results of the study. Thus, the structure of the abstract need to be changed covering, overview, problem, methods, results, conclusion.

The introduction section should be widely improved. There is unusual information that is not related with the purpose on the paper. Besides, in the introduction section the state-of-the-art overview on the problem is provided, but it is more of the type "researcher X did Y" rather than an authoritative synthesis assessing the current state-of-the-art. Where do we stand today? What seem to be the best methods/models? Have they been properly designed? Therefore, introduction should be rewritten. It is not clear which is the novelty contribution of the paper. I'd suggest to better point out what is the proposal:

-new methodology proposed by the authors for the first time.

-previously proposed methodology

-previously published papers guiding the readers to develop their own research in the field.

The authors are invited to update the introduction and refer the following reference in the literature review: Analysis of the cost for the refurbishment of small hydropower plants; Renewable Energy 34 (11), 2501-2509, 2009.

Section 2 should be moved to case study section

The Methodology in section 2 should widely improved. Thus sectionr merges theoretical part and results. It is difficult to follow with this approach

In general, result section is conducted to evaluate, analyse, or verify something. The organization of the experimentation section should be: 1) objective or aim, 2) result, 3) conclusion (outcomes and claims.) The section should be reorganized, otherwise this section is make no sense.

- The objective of the experimentation should be clearly stated at the beginning of the section.

- There are few explanations on the figures connected to section 2 . The readers can not understand what the authors want to claim. The main role of figures is to guide logic or theory of the paper, and finally prove the claim. Suitable explanations should be added.

On the case study and results: The case study is poorly presented with no sufficient data to analyse or replicate the results.

The conclusion needs to be revised. Clear contributions about this manuscript need to be presented, while also supported by some data/percentage if possible.

Author Response

The abstract changed with overview, problem, methods, results, and conclusions.

The introduction was rewritten covering with the reviewer comments.

Section 2 moved to the case study section.

The methodology part did not change too much because of the writer for this part is absent for some other reasons. I do apologize for that.

The result section added more tables, data from programming results and wrote more sentences to be clear more.

The conclusion added percentage, data, and explained more.

Thanks a lot for reviewing.

Reviewer 2 Report

General comment: This paper presents a model for minimizing water consumption. .
Introduction: The Introduction should be improve focusing more on the aim of the paper, main methods, main results and few recommendations based on empirical results. The authors should explain more their research novelty compared to previous studies from literature.
Methodology: Indicate limits and advantages of methods. Indicate alternative methods. Provide practical comments to introduce the methods.
Results: The interpretations are too superficial. More comments of the results are required and comparisons with similar studies from literature. More details on data are required.
Discussion: Interpretations of the results are provided, but a more critical position is required. The literature review should be extended.
Bibliography/References: The reference list is not up-to-date. Add recent references, especially those from journals indexed in international databases, WoS and Scopus.
Decision: Accept with corrections.

Author Response

The introduction was rewritten with the aim of the paper, recommendations based on empirical results and previous studies from the literature.

The methodology added some sentences.

The result put more tables, data and sentences to be clearly more.

The conclusion added percentage, results, and comments 

References added up to date journals.

Thanks a lot for reviewing.

Round 2

Reviewer 1 Report

The main comments that I have raised have been properly addressed by the Authors

Author Response

The main reviewer comments are already revised.

Reviewer 2 Report

More comments of the results are required and comparisons with similar studies from literature. More details on data are required.

The reference list is not up-to-date. Add recent references, especially those from journals indexed in international databases, WoS and Scopus. 

Author Response

The results added comments and compared them with other literature.

The reference added up-to-date databases.
